# On-Robot Learning With Equivariant Models

**Dian Wang**   **Mingxi Jia**   **Xupeng Zhu**   **Robin Walters**   **Robert Platt**
Khoury College of Computer Sciences
Northeastern University
Boston, MA 02115, USA
{`wang.dian,jia.ming,zhu.xup,r.walters,r.platt`}`@northeastern.edu`

**Abstract:** Recently, equivariant neural network models have been shown to improve sample efficiency for tasks in computer vision and reinforcement learning. This paper explores this idea in the context of on-robot policy learning in which a policy must be learned entirely on a physical robotic system without reference to a model, a simulator, or an offline dataset. We focus on applications of Equivariant SAC to robotic manipulation and explore a number of variations of the algorithm. Ultimately, we demonstrate the ability to learn several non-trivial manipulation tasks completely through on-robot experiences in less than an hour or two of wall clock time.

## 1   Introduction

Training directly on a physical robot is challenging because it takes a long time to gather experiences: an environmental step on a physical robot system is often at least one or two orders of magnitude slower than an environmental step in the simulation. As a result, it is not unusual for researchers who want to learn on a physical robotic system to spend hundreds of hours of robot time to learn even simple manipulation control policies, e.g. [1, 2, 3]. However, recent work has shown that policy learning using equivariant models with rotation, translation, and reflection symmetries yield much higher sample efficiency than conventional approaches [4, 5].

This paper demonstrates that with the right choices for symmetry group and data augmentation strategy, these equivariant approaches can be so sample efficient that it becomes feasible to learn simple robotic manipulation policies directly on a physical system (we call this *on-robot* learning). This newfound ability to learn simple manipulation skills quickly and directly on a robot also gives us a new perspective on the problem of the sim2real gap, the small differences between a simulation of the real world and the real world itself [6]. Our results suggest that it is not always worthwhile to train a policy in simulation first before fine-tuning in the real world, at least in the context of simple manipulation tasks where equivariant models can learn quickly.

This paper makes three contributions. First, we find that equivariance with respect to discrete symmetry groups leads to better performance than equivariance with respect to continuous groups. Although prior work [4, 5] makes it clear that equivariant policy learning can be much more sample efficient than learning with non-equivariant models, it is not clear what symmetry groups are most appropriate in robotic domains with rotation and reflection symmetries. Based on recent work, we know we can encode continuous $SO(2)$ and $O(2)$ symmetries using the irreducible representations of the group [7]. However, approximating continuous symmetry using discrete subgroups may also work well. This paper evaluates these alternatives and finds that even though the continuous group more closely reflects the actual problem symmetries, discrete rotation and reflection groups like $D_4$ and $C_8$ still have better performance.

Second, we show data augmentation further improves models equivariant to discrete groups. Since equivariant models hard code problem symmetries into the neural network model, one might assume that data augmentation would no longer be helpful. In the case of discrete groups, which only approximate the full domain symmetry, this turns out to be untrue. Here, we evaluate equivariant policy learning with and without various types of data augmentation and find that even something as simple as buffer augmentation much improves performance.

6th Conference on Robot Learning (CoRL 2022), Auckland, New Zealand.

Third, we perform a series of evaluations of on-robot learning using equivariant methods. We show that the equivariant models are so sample efficient that they can learn policies for solving various manipulation tasks from scratch within one or two hours. Furthermore, we demonstrate that sim2real pre-training is unnecessary for equivariant policy learning and is sometimes harmful. Since equivariant policy learning makes it possible to learn simple manipulation policies directly on a physical robot efficiently, it is worth asking whether the sim2real approach is still useful in these applications. We compare training exclusively on the robot with a sim2real strategy where we pre-train in a PyBullet simulation and then transfer onto the physical robot. We find – in the four representative manipulation applications explored here – that while there is often a benefit to training in simulation first before transferring to the robot, this is by no means necessary. Moreover, it is sometimes the case that the simulation and physical agents learn qualitatively different things, potentially leading to negative sim2real transfer, i.e., the situation where the pre-trained policy actually impedes learning on the physical system. Supplementary video and code are available at https://pointw.github.io/equi_robot_page/.

## 2   Related Work

**Equivariant Learning**: The first equivariant neural networks introduced were $G$-Convolution [8] and Steerable CNNs [9], which improved the sample efficiency of traditional convolutional neural networks by injecting symmetries in the structure of the neural network. Weiler and Cesa [7] proposed a framework for implementing general E(2)-Steerable CNNs. Recent work showed encouraging results for applying equivariant networks in various computer vision [10, 11] and dynamics [12, 13] tasks. They have also been applied to deep RL [14, 4] and robotic manipulation [15, 5, 16, 17] with compelling results. However, to our knowledge, equivariant methods have never been explored in the context of on-robot reinforcement learning.

**On-Robot Learning**: The most common approach to robotic policy learning is to train in simulation and then transfer to a real world application [18, 19, 20, 21, 22]. Nevertheless, there have been several efforts to develop methods that enable an agent to learn a policy directly on a physical robotic system. Gu et al. [23] trained manipulation skills in fixed environments with multiple physical robot workers. Singh et al. [24] developed a method that learned manipulation skills within 1-4 hours in the real world but required a user to respond to queries for labels. Kalashnikov et al. [2] trained a grasping policy with seven robots and over 800 robot hours. Zeng et al. [25, 26] demonstrated on-robot learning by encoding the $Q$ function using a fully convolutional network, but only in the context of open-loop tasks where the gripper performed a pre-defined behavior. FERM [27] performed on-robot learning using SAC [28] in combination with a contrastive learning objective [29, 30], but only for tasks where the orientation of the gripper was fixed. Relative to the work above, our method is most comparable to FERM [27], and we therefore benchmark our method against that.

## 3   Background

**Equivariance Over the Rotation Group:** Many robotics problems display rotational and reflectional symmetry in the plane perpendicular to gravity. These are captured by the group O(2) which contains all continuous planar rotations $\mathrm{Rot}_\theta$ about the origin and reflections through lines through the origin. It contains the subgroup of rotations $\mathrm{SO}(2) = \{\mathrm{Rot}_\theta : 0 \leq \theta < 2\pi\}$. Sometimes, we are interested in discrete subsets, for example the cyclic subgroup $C_n = \{\mathrm{Rot}_\theta : \theta \in \{\frac{2\pi i}{n} | 0 \leq i < n\}\}$ or the Dihedral group $D_n$ which contains the $n$ rotations of $C_n$ as well as $n$ reflections through $n$ evenly spaced lines through the origin. Domain symmetries can be described as invariance or equivariance of task functions. A function $f$ is *G-invariant* if when its input $x$ is transformed by a symmetry group element $g \in G$, its output stays the same, $f(gx) = f(x)$. A function $f$ is *G-equivariant* if when its input $x$ is transformed by a symmetry group element $g \in G$, its output transforms accordingly by $g$, $f(gx) = gf(x)$.

**Group Invariant MDPs:** Equivariant policy learning uses symmetries in the MDP to structure the neural network model used to represent the policy and value function. Let $M = (S, A, T, R, \gamma)$ denote an MDP and let $g \in G$ denote an element of a symmetry group $G$ (e.g., $G = \mathrm{O}(2)$). We will say that MDP $M$ is $G$-invariant if both the transition function and the reward function are invariant: $T(s, a, s') = T(gs, ga, gs')$ and $R(s, a) = R(gs, ga)$ for all $g \in G$ [5]. This type of symmetry is a

good fit for robotics problems that are invariant over rotation and reflection. In many manipulation problems, for example, the objective is to perform some task (e.g., open a drawer or insert a part) regardless of the respective poses of the parts involved.

**Equivariant SAC:** Equivariant Soft Actor Critic (Equivariant SAC) [5] is a version of SAC [28] that uses equivariant neural models to encode the symmetries of a $G$-invariant MDP. It implements the critic using an $G$-invariant model, $q(gs, ga) = q(s, a)$, and the actor using an $G$-equivariant model, $\pi(gs) = g\pi(s)$. This is illustrated in Figure 1 for a manipulation domain where $G = SO(2)$, state is an image $s = \mathcal{F}_s$, and action is a vector $a = (x, y, z, \theta, \lambda)$, where $(x, y, z)$ is the gripper position displacement, $\theta$ is the gripper orientation displacement about the $z$-axis, and $\lambda$ is the gripper aperture. The action $a$ is partitioned into $(x, y) \in A_{equi}$ and $(z, \theta, \lambda) \in A_{inv}$ so that $ga$ rotates the

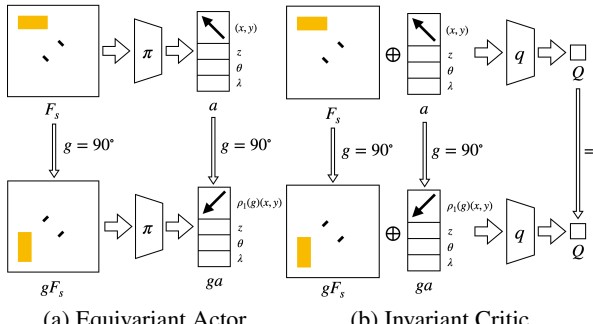

(a) Equivariant Actor    (b) Invariant Critic

Figure 1: Illustration of the Equivariant SAC. (a): the equivariant actor's output action rotates as the input state rotates. (b): the invariant critic's output doesn't change when the input state and action are rotated simultaneously.

$(x, y)$ component of action but leaves the other action dimensions unchanged. Figure 1 left shows the equivariance of the actor. When the state image rotates by 90 degrees, the $x, y$ components of action rotate but the $z, \theta, \lambda$ components remain unchanged. The right side shows the invariance of the critic. Corresponding rotations of state and action do not change the output $Q$ value.

## 4  Symmetry Group and Augmentation Strategy

The specific choice of symmetry group and data augmentation strategy have a significant impact on the sample efficiency of the algorithm. Before evaluating our algorithms on the robot, we evaluate those different algorithmic choices in simulation. Here, we experiment in the context of Equivariant SAC for the four tabletop manipulation tasks shown in Figure 2.

All tasks have sparse rewards, i.e., +1 reward for reaching the goal, and 0 otherwise. We use a 2-channel image as the observation. The first channel is a top-down depth image centered with respect to the robot gripper. The gripper is drawn at the center of the depth image with its current aperture and orientation. The second channel is a binary channel (i.e., the values of all pixels are either 0 or 1), indicating if the gripper is holding an object. The action space is: $x, y, z \in [-0.05m, 0.05m]; \theta \in [-\frac{\pi}{4}, \frac{\pi}{4}]; \lambda \in [0, 1]$ (0 means fully close and 1 means fully open). 20 episodes of expert demonstration are added to the replay buffer before the start of training (see the ablation study about the effect of expert demonstrations in Appendix G). See the detailed description of the environments in Appendix B and the training details in Appendix D.

### 4.1  Choice of Symmetry Group

In Equivariant SAC, we must select a symmetry group with which to parameterize the actor and critic models. This symmetry group reflects the underlying symmetry that we assume to exist in the problem. One might expect that larger symmetry groups would always be better because they would enable the model to generalize to a greater number of different situations. However, as shown in Weiler and Cesa [7], a larger group does not necessarily improve the performance of the model. It is therefore important to ask which symmetry groups are most helpful in our robotic manipulation domains. We compare the performance of Equivariant SAC when the actor/critic models are parameterized by each of the following five different symmetry groups: 1) $C_8$: the cyclic group that encodes discrete rotations every 45 degrees; 2) $D_4$: the Dihedral group that encodes rotations every 90 degrees and reflection; 3) $D_8$: the Dihedral group that encodes rotations every 45 degrees and reflection; 4) $SO(2)$: the group of continuous planar rotations; 5) $O(2)$: the group of continuous planar rotations and reflections. In all baselines, we use data augmentation in the replay buffer where four extra transitions with random $SO(2)$ is generated for each experienced transition.

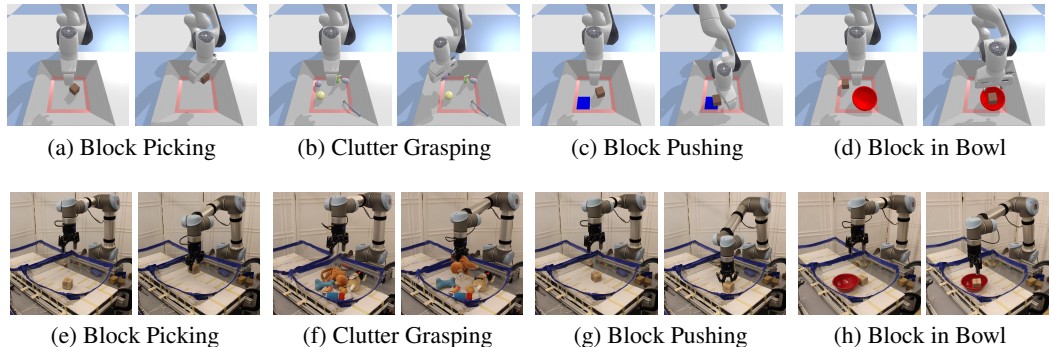

| (a) Block Picking | (b) Clutter Grasping | (c) Block Pushing | (d) Block in Bowl |
|---|---|---|---|

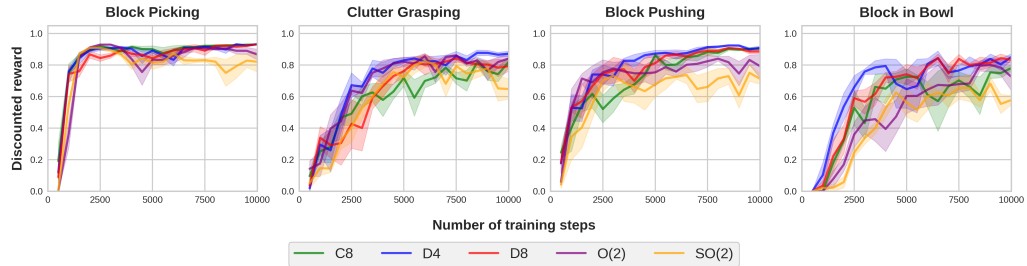

| (e) Block Picking | (f) Clutter Grasping | (g) Block Pushing | (h) Block in Bowl |
|---|---|---|---|

Figure 2: (a)-(d): Our simulation environments implemented in PyBullet [31]. The left images in each environment show the initial state of the environment; the right images in each environment show the goal state. (e)-(h): Our on-robot learning environments.

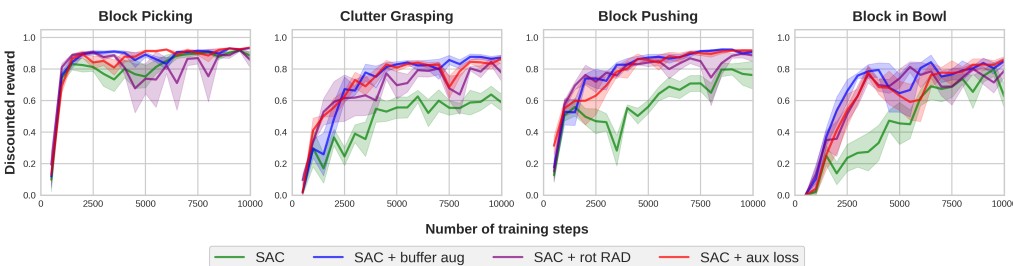

(a) Comparison of Equivariant SAC defined with different symmetry groups.

(b) Comparison of Equivariant SAC equipped with different data augmentation techniques.

Figure 3: The plots show the performance of the behavior policy in terms of the discounted reward. Each point is the average discounted reward in the previous 500 steps. Results are averaged over four runs. Shading denotes standard error.

Figure 3a shows the result. In all four environments, $D_4$ and $D_8$ show the best performance over the five different groups in terms of convergence speed and converged performance, where $D_4$ has a marginal improvement over $D_8$ even though $D_8$ encodes more rotations than $D_4$. In addition, we find that incorporating reflection symmetry generally improves the performance (comparing $D_4$ vs $C_8$ and O(2) vs SO(2)). However, both the SO(2) and O(2) networks underperform the $D_4$ network. We hypothesize that this is because the $D_4$ network has access to the regular representation of the symmetry group as the hidden layer of the network, whereas the models defined over the continuous groups SO(2) and O(2) do not. The regular representation explicitly encodes the feature maps over all elements in the finite group concurrently, making it an informative representation for the hidden layers. Moreover, the regular representation is compatible with component-wise activation functions and component-wise max pooling, which are most commonly used and known to work well in deep networks.

## 4.2  Choice of Data Augmentation Strategy

The results of Section 4.1 indicate that the equivariant models that perform best in our domains use the $D_4 \subset O(2)$ symmetry group, the group of 90 degree rotations and reflections. As a result, it could still make sense to use data augmentation to enable the model to learn rotational symmetry within a continuous range $(0, \frac{\pi}{2})$. Here, we compare three different approaches to data augmentation with random rotations: SAC + buffer aug, SAC + aux loss, and SAC + rot RAD. In *SAC + buffer aug*, we add four extra transitions with random $SO(2)$ rotations to the replay buffer for each experienced transition. (See Appendix F for an comparison of different amounts of augmentation.) In *SAC + aux loss*, we add extra loss terms to encourage the model to learn continuous rotational equivariant representations: $\mathcal{L}_{\text{aux}}^{\text{actor}} = \frac{1}{2}(\pi(g\mathcal{F}_s) - g\pi(\mathcal{F}_s))^2$ and $\mathcal{L}_{\text{aux}}^{\text{critic}} = \frac{1}{2}(q(g\mathcal{F}_s, ga) - q(\mathcal{F}_s, a))^2$, where $g \in SO(2)$ and $(\mathcal{F}_s, a)$ is the sampled state-action pair in the minibatch. In *SAC + rot RAD*, we use RAD [32] to perform rotational data augmentation at each sample and training step. Figure 3b shows the comparison between the three data augmentation approaches and vanilla Equivariant SAC (green). Even though Equivariant SAC already encodes D4 equivariance within the structure of the network, adding $SO(2)$ data augmentations to the algorithm can still help by a substantial margin, especially in the more challenging tasks like clutter grasping, block pushing, and block in bowl. However, notice that all three data augmentations approaches perform similarly. As such, we adopt the buffer augmentation method in the remainder of this paper because it is the simplest method.

## 5  On-Robot Learning

Figure 4 illustrates our on-robot learning setup. We use a Universal Robots UR5 arm equipped with a Robotiq 2F-85 parallel-jaw gripper. Since we need to be able to see the objects on the table beneath the robot hand and arm, we mount two depth sensors that view the scene from orthogonal directions each at 45 degrees with respect to the bins (see Figure 4 left). One of these sensors is an Occipital Structure Sensor and the other is a Microsoft Azure Kinect DK. The output of each sensor is con-

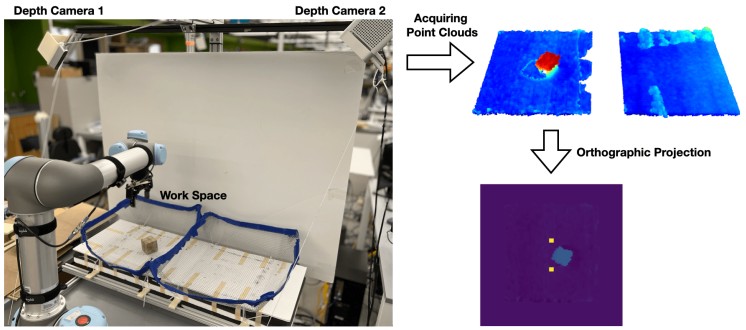

Figure 4: Our experimental set up for on-robot learning. The observation (bottom right) is generated by first acquiring point clouds from two depth cameras above the workspace then creating an orthographic projection at the gripper's position. The gripper is drawn at the center of the observation (in yellow) with its current aperture and orientation.

verted to a partial point cloud, fused into a single combined point cloud, and projected into a depth image viewed from a top-down direction (Figure 4 right). We use the same 2-channel observation as in Section 4 including a depth channel and a binary channel for the gripper state. Our workstation has a Intel Core i7-9700k CPU (3.60GHz) and a Nvidia RTX 2080Ti GPU. All together, this setup enables us to train at a rate of approximately 1.1 seconds per environmental transition. This includes the time it takes to take a single transition on the physical robot as well as to perform a single SGD step in the model (we do this in parallel with the robot motion), but it does not include the time it takes to reset the environment between episodes.

### 5.1  Physical Tasks

All physical experiments are performed using the same four tasks described in Section 4 (see Figure 2) [1]. In tasks involving only a single object (i.e., Block Picking and Block Pushing), the robot uses only one bin as the workspace. In tasks involving multiple objects (i.e., Clutter Grasping and Block in Bowl), the robot alternately uses one of the two bins as the active workspace and then uses

---

[1]Note that in Block Pushing, we use a virtual goal drawn on the observation, so the goal is visible to the agent but not to the human observer.

| Task | Block Picking | Clutter Grasping | Block Pushing | Block in Bowl |
|---|---|---|---|---|
| Number of training steps | 2000 | 2000 | 2000 | 4000 |
| Approximate time for training | 45 mins | 45 mins | 1 hr | 2 hrs 40 mins |
| Evaluation success rate | 100% (50/50) | 96% (48/50) | 92% (46/50) | 92% (46/50) |
| Sim2real transfer success rate | 100%(50/50) | 86%(43/50) | 70%(35/50) | 72%(36/50) |

Table 1: Top: the number of training steps, approximate time for training, and the evaluation success rate of the trained policy of our on-robot learning. Bottom: the evaluation success rate of the model trained in simulation. All succes rates are averaged over 50 episodes.

the other bin to reset the environment. We implement an automated resetting process for all four environments. In Block Picking and Block Pushing, the robot will reset the environment through picking up the block and randomly placing it inside the workspace. In Block in Bowl, the robot will move both the bowl and the block from the active workspace to random positions in the reset bin, then switch the active workspace to the reset bin. In Clutter Grasping, after the robot successfully grasps one object from the active workspace, it will drop the object to a random position in the reset bin. Once the robot grasps all objects from the active workspace, the robot will switch the active workspace to the reset bin. See Appendix C for more details.

## 5.2 Performance of the On-Robot Learned Policy

First, we evaluate the ability of Equivariant SAC to learn these tasks entirely in the on-robot setting, i.e., entirely on the robot without pre-training. Based on the findings described in Section 4, we configure both our actor and critic models to use the $D_4$ symmetry group, and we use the "SAC + buffer aug" strategy described in Section 4.2. The blue line in Figure 5 shows the learning curve of training, and the third row of Table 1 shows the greedy performance of the learned policy after training, averaged over 50 test episodes.

In Block Picking, the robot succeeded in all 50 trials. In Clutter Grasping, the robot failed to find an appropriate grasp point in two out of the 50 episodes. In Block Pushing, the robot failed in four out of 50 episodes. In two of these, it failed to move towards the block. In the other two failures, the robot kept pushing down from the top of the block and triggered the UR5 safety system. In Block in Bowl, the robot failed in four out of the 50 trials. In three of these, the robot grasped the block but did not move towards the bowl. In the other failure, the robot grasped the bowl and the block together in a single grasp and failed to let go.

In Block Picking, Clutter Grasping, and Block Pushing, training lasts 2000 transitions. In Block in Bowl, training lasts 4000 transitions because it is a harder task to learn. In terms of wall clock time, training tasks a total of 45 minutes in Block Picking and Clutter Grasping; one hour in Block Pushing (because of the additional time required to physically reset that environment); and 2 hours and 40 minutes in Block in Bowl. For Block in Bowl, this time can be decomposed into approximately 73 minutes ($1.1s \times 4000$) of environmental steps and 107 minutes to reset the environment.

## 5.3 Baseline Comparison

There are very few methods in the literature that can learn efficiently in the on-robot setting. Here, we baseline against the Framework for Efficient Robotic Manipulation (FERM) [27]. FERM utilizes contrastive learning with random crop augmentations to improve sample efficiency. We implement FERM such that the image encoder has a similar amount of trainable parameters as the equivariant network (1.5M vs 1.1M). See Appendix E for details. Figure 5 compares the learning curves of Equivariant SAC (blue) and FERM (red) where both methods learn on-robot. In Block Picking, Equivariant SAC masters the task after about 1000 steps, while FERM learns much more slowly. In Clutter Grasping, FERM performs better at the beginning phase of learning but fails to converge to a good policy at the end of learning. We hypothesize that this is because: 1) the pre-training of the encoder in FERM helps the network to learn a better feature representation at the early phase of learning. 2) this task does not require accurate manipulation as the other three tasks since the objects are all deformable (this also explains why the on-robot learning is faster than the simulation

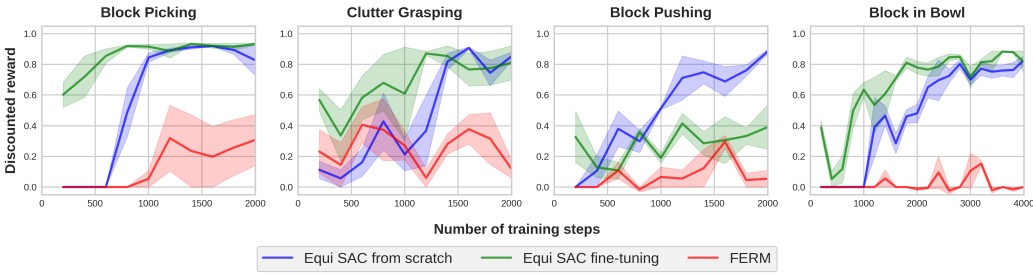

Figure 5: Comparison of Equivariant SAC trained from scratch (blue), Equivariant SAC with sim-to-real fine-tuning (green), and FERM (red) in real world. The plots show the performance of the behavior policy in terms of the discounted reward. Each point is the average discounted reward in the previous 200 steps. Results are averaged over three runs. Shading denotes standard error.

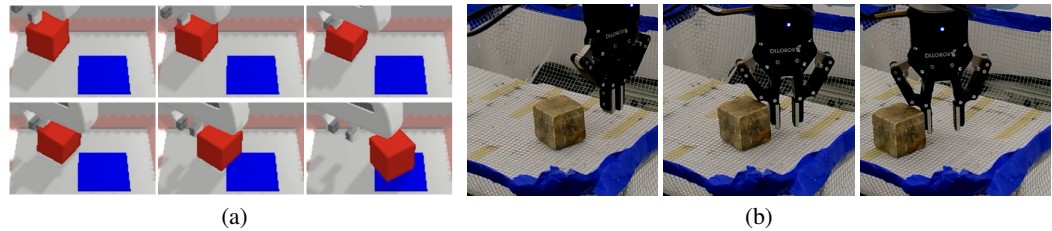

Figure 6: The different strategy between the simulation agent and the on-robot learning agent. (a): In simulation, the agent presses on the edge of the block and bounces it away. (b): In the real world, the agent pushes the block.

in this task). In Block Pushing and Block in Bowl, FERM fails to learn any good policy at all, while Equivariant SAC solves the task within 2000 and 4000 steps, respectively.

## 5.4 Sim2real Comparison

Here, we compare direct on-robot learning with a sim2real strategy where we first train the agent to convergence in simulation and then transfer to the physical system by directly copying model parameters. We used the PyBullet simulation environment described in Section 4. The last row of Table 1 shows the performance of the agent trained in simulation when that policy is executed on the physical robot. In general, the sim2real policy performs significantly worse than the on-robot trained policy: 86% versus 96% in Clutter Grasping, 70% versus 92% in Block Pushing, and 72% versus 92% in Block in Bowl. The exception is Block Picking in which both the sim2real and on-robot policies achieve 100% success. This result explicitly reflects the sim2real performance gap.

## 5.5 Fine-tuning Sim2real on Physical Robots

We also evaluated our ability to improve the policy learned in simulation by fine-tuning on the physical robot. The green line in Figure 5 shows the learning curve of this fine-tuning agent. Before fine-tuning begins, the models used by this agent are loaded with the model parameters learned in simulation. This should be compared with the performance of the on-robot agent (no pre-training), shown in blue in Figure 5. In Block Picking, Clutter Grasping and Block in Bowl, both the training from scratch and sim2real fine-tuning converge to the same level of performance at the end of training, where the training from scratch agent is about 500 steps slower than the fine-tuning agent in terms of converging speed. This suggests that while pre-training helps in these tasks, it is not critical to successfully learning the policy.

In Block Pushing, on the other hand, the fine-tuning agent performs very poorly and it does not recover during training (green versus blue line in Figure 5). To understand why this occurs, we checked the policies typically learned by the different agents for Block Pushing. Whereas the sim-

ulation agent learns to move the block by pushing down on the block *from the top* (see Figure 6 (a)), the on-robot agent learns a policy that pushes the block *from the side* (see Figure 6 (b)). We hypothesize that this is what impedes the sim2real fine-tuning agent on this task. In order to succeed on the physical robot, the sim2real agent must first *unlearn* the policy that was successful in simulation. This is an example of negative transfer where the pre-trained policy actually impedes on-robot learning. The fact that the simulation agent learns to push down on the block (and then to push sideways) whereas the on-robot agent learns to push from the side is probably the result of differences between the contact dynamics of the simulator and the real robot. The PyBullet simulator has slightly softer contact compliance that enables it to push down without generating too much force and triggering the UR5 safety system. Also, the friction coefficient between the gripper and the block is larger than between the block and the ground, allowing the simulator to slide the block. Notice that it would be very hard to close this gap by improving the simulator because it is difficult to measure and model physical friction and compliance accurately.

## 6 Discussion

This paper proposes an approach to on-robot learning using Equivariant SAC in combination with data augmentation that can learn to solve simple manipulation tasks in a couple of hours without pre-training. Using the new method, we find that it may sometimes be unnecessary to pre-train in simulation before training on a physical robot and that pre-training in simulation can sometimes be harmful because it causes negative transfer.

## 7 Limitations

A key limitation of the approach is that the use of equivariant models requires making assumptions about which symmetries are present in the domain. While many robotics problems have rotation, translation, and reflection symmetries, these symmetries may not be present in all regions of the state space. The current approach requires the system designer to model these asymmetries explicitly. However, the system would ideally have the ability to learn where symmetries are present without hand coding. Similarly, many problems have domain-specific symmetries beyond rotation and reflection symmetries that can be hard for the system designer to recognize. We would like our system to be able to identify which symmetries are present and then incorporate them into the relevant neural models. Another area for future work is to understand when incorrect symmetry assumptions are still useful. We have observed that equivariant models can still speed up learning, even when the symmetry assumptions are sometimes violated in a specific domain. We would like to have a theory for understanding this phenomenon more precisely. Third, though our method is not limited to depth images, this work demonstrates equivariant learning using depth images only because our tasks do not require RGB information. In the future work, we will incorporate RGB image to improve the robustness of our method to transparent or reflective objects.

## Acknowledgments

This work is supported in part by NSF 1724257, NSF 1724191, NSF 1763878, NSF 1750649, and NASA 80NSSC19K1474. R. Walters is supported by the Roux Institute and the Harold Alfond Foundation and NSF grants 2107256 and 2134178.

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
