# OpenReview forum: "On-Robot Learning With Equivariant Models"
_robot-learning.org/CoRL/2022/Conference — CoRL 2022 Poster_

### Official Review · Reviewer_vBAF · 2022-07-27

**Originality:** Good
**Technical Quality:** Good
**Clarity Of Presentation:** Very Good
**Impact:** 3

**Recommendation:**

Weak Accept: I recommend accepting the paper, but will not argue for my recommendation if the majority of other reviewers have a different opinion.

**Summary:**

This paper proposes an on-robot equivariant model to handle manipulation tasks. The work leverages the task structures such as symmetry explicitly for efficient learning of a control policy.  Moreover, the paper finds that data augmentation is still a powerful tool together with equivariant models. The experiments demonstrate the effectiveness of the work as well as compare different types of equivariant models.

**Issues:**

The key issue is that the method might not be scalable to different tasks or camera settings.
The lack of in-depth analysis of data augmentation in equivariant models. The reason why it works is not clear.

**Quality Of The Limitations Section:**

Additional details required

**Reviewer Expertise:**

4: The reviewer is confident but not absolutely certain that the evaluation is correct

**Robotics Focus:**

Sufficient demonstration on hardware

**Strengths And Weaknesses:**

Strengths:
The paper is clearly written.
The paper demonstrates the effectiveness of equivariant models for on-robot learning. While the idea itself is not new, the overall system is valuable.
The authors carefully study the variants of different equivariant models. This shed light on future research in this direction.

Weaknesses:
The limitation of this work stands out. The requirement of the task structure prevent the method from scaling up to a general method.
The tasks are relatively on the easy end of the family. But this is a minor issue.
The robustness of the proposed method is not proved. For example, would this method work with arbitrary camera angle？


**Summary Of Recommendation:**

This work aligns well with the scope of CORL. It demonstrates the effectiveness of equivariant models on real robots. The authors test with various implementations and provide useful information for future works.

---

> ### Author Response · Authors · 2022-08-23
> **Author Response to Reviewer vBAF**
>
> The authors thank the reviewer for their careful review. Please see our comments below.
>
> >The requirement of the task structure prevent the method from scaling up to a general method. The tasks are relatively on the easy end of the family. But this is a minor issue. The robustness of the proposed method is not proved. For example, would this method work with arbitrary camera angle?
>
> This is a good point, our current approach does require a fully-equivariant setting (e.g., a top-down image is required). In our ongoing future work, we have some preliminary results demonstrating that variations of this method work well in partially equivariant problem settings where an arbitrary camera angle is used. However, we feel that learning on a robot in even the fully equivariant settings is important.
>
> >The lack of in-depth analysis of data augmentation in equivariant models. The reason why it works is not clear.
>
> It is very common to use data augmentation in equivariant models [7, 8, 9]. Data augmentation typically helps the equivariant networks as long as the augmentation comes from a group larger than that the equivariant network encodes [8]. As we point out in section 4.2., the network only enforces equivariance in $D_4 \subset O(2)$, and using data augmentation helps the model to learn the rotational symmetry in the continuous range of $(0, \frac{\pi}{2})$.

---

### Official Review · Reviewer_MYEX · 2022-07-29

**Originality:** Good
**Technical Quality:** Good
**Clarity Of Presentation:** Very Good
**Impact:** 3

**Recommendation:**

Weak Accept: I recommend accepting the paper, but will not argue for my recommendation if the majority of other reviewers have a different opinion.

**Summary:**

The paper presents a study on using equivariant models for deep reinforcement learning of robotic manipulation policies. It studies a variant of a well-known deep RL algorithm, soft actor-critic, when using rotation and reflection equivariant neural architectures for the learned components, as in prior work on equivariant SAC. The authors experiment to determine which symmetry groups lead to improved performance, and also show that interesting, further data augmentation can improve sample efficiency on top of discrete equivariances. They use real-world robot learning experiments to demonstrate the sample efficiency afforded by equivariant models.

**Issues:**

- I think it would be helpful to see the performance of non-equivariant SAC with the expert demonstrations in the buffer as well as non-equivariant SACfD, both in the simulated and real experiments. The claim in the paper is made that the sample efficiency of the equivariant method is significantly improved, but it’s never shown in any experiments. I would agree that in general on-robot learning using RL from scratch can be extremely time-consuming, but it would depend on the action space, reward, expert demonstrations provided, etc. so it would be good to be able to see that comparison here as well.
- Why were 20 expert demonstrations chosen for the real-robot experiments? It seems from the experimental results in Figure 13 that 10 demonstrations doesn’t seem significantly worse.
- Is the input to the networks in the real robot experiments a depth image, or is it a two-channel image like in the simulated experiments which also includes the binary channel indicating whether the gripper is holding the object?
- It would be helpful for clarity to include that the plots in Figure 3(a) are generated using buffer augmentation as the additional data augmentation strategy (if I am not mistaken).



**Quality Of The Limitations Section:**

Limitations are addressed clearly

**Reviewer Expertise:**

3: The reviewer is fairly confident that the evaluation is correct

**Robotics Focus:**

Sufficient demonstration on hardware

**Strengths And Weaknesses:**

Strengths:
 - The motivation is very strong – sample efficiency is a significant limiting factor for current deep RL algorithms when trying to learn robot control policies from scratch.
- The authors are able to successfully demonstrate real world on-robot learning with impressive sample efficiency, which is quite a nice result.
- The experimental contributions of the paper are quite useful for practitioners and I think will be of interest to the community.

Weaknesses:
- The methods evaluated in the paper have limited novelty, and are mostly drawn from prior work. However, the experiments and analysis of these methods still provide significant value.
- The first claim is that $D_4$ is empirically the best symmetry group is hard to conclude given the results in Figure 3, since the error bars are quite wide. Also, the statement that “In all four environments, $D_4$ shows the best performance over the four different groups” seems unclear to me – which metric is this based on? For example, $D_8$ does not seem to be worse than $D_4$ for the “block in bowl” task.
- As with all equivariant methods, much care must be taken to ensure that the action and state space satisfy the conditions necessary for invariance and equivariance. This limits the types of tasks that such methods can be applied to. In spite of this, I believe that many important robotic manipulation tasks can be framed in such a way.
- The reset mechanism for the block picking task requires the robot to already be able to pick up the block which somewhat obviates the task, but this could have been done using a number of other reset mechanisms (which aren’t the focus of this work).


**Summary Of Recommendation:**

This paper presents a nice empirical study on using methods like equivariant SAC for real robot learning. It is a good fit for the CoRL venue and I think it is well motivated and interesting. Some of the claims made in the paper do not seem to be backed by experimental results with significant effect size, which is my main concern. However, I think the demonstration of highly sample efficient on-robot deep model-free RL is quite impressive. I currently vote for weak acceptance.

---

> ### Author Response · Authors · 2022-08-23
> **Author Response to Reviewer MYEX**
>
> The authors thank the reviewer for their careful review. Please see our comments below.
>
> >The first claim is that D_4 is empirically the best symmetry group is hard to conclude given the results in Figure 3, since the error bars are quite wide. Also, the statement that “In all four environments, D_4 shows the best performance over the four different groups” seems unclear to me – which metric is this based on? For example, D_8 does not seem to be worse than D_4 for the “block in bowl” task.
>
> The metrics are convergence speed and converged performance. In all four environments, $D_4$ either converges faster than other groups and/or converges to higher performance. In Block in Bowl, $D_4$ and $D_8$ converge to similar performance, but $D_4$ learns much faster than $D_8$. However, we agree that the outperformance of $D_4$ compared with $D_8$ is marginal, and we revised our statement in the revision.
>
> >I think it would be helpful to see the performance of non-equivariant SAC with the expert demonstrations in the buffer as well as non-equivariant SACfD, both in the simulated and real experiments. The claim in the paper is made that the sample efficiency of the equivariant method is significantly improved, but it’s never shown in any experiments. I would agree that in general on-robot learning using RL from scratch can be extremely time-consuming, but it would depend on the action space, reward, expert demonstrations provided, etc. so it would be good to be able to see that comparison here as well.
>
> Thanks for pointing this out, we added those comparisons in simulation in the revision (Appendix H, Fig 15 and 16). As for a comparison in the real-world experiment, FERM is a state-of-the-art non-equivariant method that we have already compared against.
>
> >Why were 20 expert demonstrations chosen for the real-robot experiments? It seems from the experimental results in Figure 13 that 10 demonstrations doesn’t seem significantly worse.
>
> We select 20 because of the underperformance of 10 demonstrations in Block Pushing and Block in Bowl in Figure 13, and we want the number of demonstrations to be the same for all four environments.
>
> >Is the input to the networks in the real robot experiments a depth image, or is it a two-channel image like in the simulated experiments which also includes the binary channel indicating whether the gripper is holding the object?
>
> The input is the two-channel image like in the simulation experiments. We clarified this in the revision.
>
> >It would be helpful for clarity to include that the plots in Figure 3(a) are generated using buffer augmentation as the additional data augmentation strategy (if I am not mistaken).
>
> Thanks for pointing this out. You are right, we clarified this in the revision.

---

> > ### Comment · Reviewer_MYEX · 2022-08-26
> > **Response to authors**
> >
> > Thank you for your detailed responses to my questions. Most of my questions have been clarified, and I continue to vote for acceptance.

---

### Official Review · Reviewer_QgU1 · 2022-08-01

**Originality:** Fair
**Technical Quality:** Good
**Clarity Of Presentation:** Fair
**Impact:** 3

**Recommendation:**

Weak Accept: I recommend accepting the paper, but will not argue for my recommendation if the majority of other reviewers have a different opinion.

**Summary:**

The paper applies the recent progress of equivariant policy learning to top-down grasping using trained on real robot arms. The experiments show that the effectiveness of high sample efficiency of the equivariant policy learning.


**Issues:**

1. More details about what D_n and C_n groups mean in the context of depth images
2. Discussion about why using depth and why not using RGB images. Is it simply because depth images give better sample efficiency, or there is some difficulties to prevent RGB images from being using by Equivariant SAC?
3. Clearly state this work is based on Equivariant SAC (SO(2)-equivariant reinforcement learning) and E2CNN (General E(2)-Equivariant Steerable CNNs).

**Quality Of The Limitations Section:**

Additional details required

**Reviewer Expertise:**

4: The reviewer is confident but not absolutely certain that the evaluation is correct

**Robotics Focus:**

Sufficient demonstration on hardware

**Strengths And Weaknesses:**

## Strengths

The on-robot results are pretty impessive in terms of the sample-efficiency of learning process. The results are compared to one of the state-of-the-art methods FERM (Framework for Efficient Robotic Manipulation) in top-down grasping leanring, and show superior performance. In addition to purely on-robot learning bootstrapped from 20 human demonstrations, sim2real results are shown and analyzed. It's also worth-noting that this paper discussed the limitations of using equiviance in policy learning very well in the limitation section.

## Weakness

1. The delta of this paper's methodology from the prior work, Equivariant SAC (SO(2)-equivariant reinforcement learning), that this paper is largely based on, is small. Indeed, authors discovered that using discrete rotation and reflection groups for equivariant sac training can lead to better performance than the continuous counterpart, and found some interesting phenomena in sim2real transfer. However, the change in equivariant groups doesn't seems like a crucial part of the success of fast on-robot learning, since Figure 3 (a) only demonstrates minor performance boost by not using SO(2).

2. Similar to the prior work Equivariant SAC, depth images are used as input to the policy. One limitation of using depth images is that they are not robust to many objects (e.g. plastic bottle, reflective silverwares) that are common in daily life. One solution to this is to directly use RGB images like what FERM did. It will be interesting to see a discussion about why not using RGB for equivariant policy learning.

**Summary Of Recommendation:**

Although the methodology of is this work is largely based on the prior work (Equivariant SAC), I would still recommend for a weak accept. I appreciate the large amount of work done by authors to make on-robot learning work really well, which aligns with this conference by creating learning methods that actually work on real robots and real problems.

---

> ### Author Response · Authors · 2022-08-23
> **Author Response to Reviewer QgU1**
>
> The authors thank the reviewer for their careful review. Please see our comments below.
>
> >Similar to the prior work Equivariant SAC, depth images are used as input to the policy. One limitation of using depth images is that they are not robust to many objects (e.g. plastic bottle, reflective silverwares) that are common in daily life. One solution to this is to directly use RGB images like what FERM did. It will be interesting to see a discussion about why not using RGB for equivariant policy learning.
>
> This is a good point; thanks for pointing this out. Our algorithm is not limited to depth images. One can use the same algorithm but change the input to an RGB or RGBD image. The main reason that we only use depth images is that: 1) it is sufficient in our current tasks; 2) it is easier to reconstruct a depth image from two sensors than to reconstruct an RGBD image. However, we do agree that depth images have their limitations. In fact, in our ongoing future work, we are using RGBD images. We added a discussion about only using the depth images in the limitation section.
>
> >More details about what D_n and C_n groups mean in the context of depth images
>
> $C_n$ is a group containing n rotations evenly distributed in the $0$ to $2\pi$ range. $D_n$ includes $C_n$ but has n additional reflections. In the context of images, a group element will rotate or reflect the image.

---

### Author Response · Authors · 2022-08-23
**Summary of Revision**

**Comment:**

The authors thank all reviewers and the area chair for their careful review. We have edited the paper in the revision based on the reviewer’s suggestions to strengthen our paper. We have uploaded the revision with the changes labeled in blue. Please see a summary of the changes below:
1. Clarified that the experiment in Section 4.1 also uses data augmentation in the replay buffer.
2. Revised the conclusion in Section 4.1 regarding $D_4$ only having a marginal outperformance compared with $D_8$.
3. Clarified that the experiment in Section 5 uses the same 2-channel observation as in Section 4.
4. Added a discussion about only using depth images in the limitation section.
5. Add two new experiments in Appendix H comparing Equivariant SAC (and Equivariant SACfD) with various CNN baselines.


**Zip File:**

/attachment/3fa658ae980aaeae42d3c9442dfcccd129f09c11.zip

---

### Meta-Review · Area_Chair_odcR · 2022-08-02

**Recommendation:** Accept (Poster)
**Confidence:** 4

**Metareview:**

Strengths:

+ Extensive set of experiments demonstrate impressive performance on real robots, something which is generally lacking and thus highly valued in the related literature.
+ The paper is clear and well written.

Weaknesses:

- The paper does not present any algorithmic and technical advances compared to previous work on equivariant learning, but rather empirical findings from an extensive set of experiments on real robots. Nevertheless, this has real value for the community.
- Some of the conclusions are insufficiently supported by data, since some of the stated differences in performance between some of the experimental conditions are in fact marginal and inside the noise margins of a typical learning curve. This downside is somewhat alleviated by the fact that experiments show averages over multiple seeds. Still, conclusions can be drawn on only a subset of the presented tasks.

**Best Paper Nomination:**

No

---

> ### Author Response · Authors · 2022-08-23
> **Author Response to Area Chair odcR**
>
> The authors thank the area chair for their summary. Please see our comments below.
> >The paper does not present any algorithmic and technical advances compared to previous work on equivariant learning, but rather empirical findings from an extensive set of experiments on real robots.
>
> We agree that there is no significant algorithmic contribution in this paper. However, we feel that our demonstration that equivariant RL can be used directly on a physical robot is important enough to warrant a paper. Our paper also evaluates some parameter alternatives w.r.t. the physical robotics setting.
>
>
> >Some of the conclusions are insufficiently supported by data, since some of the stated ifferences in performance between some of the experimental conditions are in fact marginal and inside the noise margins of a typical learning curve. Based on a single seed, one can not conclude the superiority of one condition over another based on such small differences.
>
> First, we would like to point out that our experiments are not based on a single seed. As is noted in the figure captions, all of our simulation experiments are averaged over four seeds and all of our real-world experiments are averaged over three seeds. Second, we do agree that some (but not all) of the differences in performance between the experimental conditions are marginal. In particular, we clarified that $D_4$ only has a marginal improvement over $D_8$ in the revision. However, most of our main conclusions are drawn from the large performance differences (e.g., incorporating reflection symmetry improves the performance in Fig. 3(a), and incorporating data augmentation improves the performance in Fig. 3(b)).